# Could MRONJ Be Related to Osimertinib Monotherapy in Lung Cancer Patients after Denosumab Suspension?

**DOI:** 10.3390/healthcare12040457

**Published:** 2024-02-10

**Authors:** Marta Forte, Antonio d’Amati, Luisa Limongelli, Massimo Corsalini, Gianfranco Favia, Giuseppe Ingravallo, Giuseppe Barile, Saverio Capodiferro

**Affiliations:** 1Department of Interdisciplinary Medicine, University of Bari, Piazza Giulio Cesare, 11, 70124 Bari, Italy; fmarta@live.it (M.F.); luisanna.limongelli@gmail.com (L.L.); massimo.corsalini@uniba.it (M.C.); gianfranco.favia@uniba.it (G.F.); saverio.capodiferro@uniba.it (S.C.); 2Unit of Anatomical Pathology, Department of Precision and Regenerative Medicine and Ionian Area, University of Bari, Piazza Giulio Cesare, 11, 70124 Bari, Italy; giuseppe.ingravallo@uniba.it

**Keywords:** MRONJ, Denosumab, Osimertinib, lung cancer, prevention, jaws, oral cavity

## Abstract

Background: Medication-related osteonecrosis of the jaws is the most frequent complication in patients treated or in therapy with antiresorptive/antiangiogenetic drugs. The list of medications possibly related to MRONJ onset is constantly growing; we aimed to report on a third-generation epidermal growth factor receptor tyrosine kinase inhibitor (Osimertinib) as possibly responsible for bilateral maxillary necrosis onset in the herein-described case. Methods: In June 2023, an oncologic patient with two different maxillary bone exposures was referred to our attention. His medical history revealed a two-year Denosumab regimen along with Osimertinib, the latter not suspended before teeth extractions. The clinicians performed a sequestrum removal and bone debridement after three cycles of antibiotic therapy. Results: Histologic examinations confirmed the clinical diagnosis of MRONJ excluding a metastatic occurrence, while complete mucosal healing was achieved after 15 days. Conclusions: The patient suspended Denosumab for more than six months before teeth extraction for MRONJ prevention; hence, failure to discontinue Osimertinib led us to consider it a possible etiological factor. From a literature analysis, only one case has already been published reporting a possible Osimertinib-related occurrence of MRONJ in lung cancer patients. Our case is a further report that could be intended as an alert both for oncologists and dentists to share decisions about the oral management of such patients together, also informing them about this possible risk. Also, this report could trigger in the scientific community the necessity to evaluate further guidelines for similar doubtful cases in which the drug interaction, the mono-suspension, and the possible removable prosthesis-related additional trauma should be considered causes or con-causes.

## 1. Introduction

Medication-related osteonecrosis of the jaws (MRONJ) represents a severe drug-related complication occurring in patients treated or in treatment with bisphosphonates (BPs), Denosumab, angiogenesis inhibitors, or other antiresorptive medications used for metabolic bone diseases, osteoporosis, or several malignancies such as bone metastases of solid tumors (breast, prostate, and lung cancer) and multiple myeloma, and for the adjuvant therapy of pathologic fractures [1,2].

MRONJ has been clinically defined by the “American Association of Oral and Maxillofacial Surgeons” in the updated position paper (published in 2022) [1] as an “*exposed bone or bone that can be probed through an intraoral or extraoral fistula(e) in the maxillofacial region that has persisted for more than 8 weeks, in patients with no history of radiation therapy to the jaws or metastatic diseases to the jaws, and in current or previous treatment with antiresorptive therapy alone or in combination with immune modulators or antiangiogenic medications*”. On such bases, the clinical onset and clinical presentation of MRONJ suffer from extremely high variability because the actual pathogenesis remains debated and incompletely understood.

Different critical molecular events have been associated with the genesis of MRONJ, summarized as follows. In healthy individuals, the bone addresses the infectious and inflammatory stimulus with osteoclast activation and progressive resorption [3]. The removal of the infective pathogenic noxa leads to soft and hard tissue healing due to osteoblast stimulation. In patients who begin taking medications that variably interfere with/alter the normal osseous metabolism (and consequently may be related to jaw osteonecrosis onset), there is serious osteoclast and vascular inhibition; as such, the alveolar bone provides an ineffective response to the infective and inflammatory stimulus, thus resulting in bone necrosis [4]. In particular, osteonecrosis and osteocyte death enhance the production of pro-inflammatory cytokines such as IL-17, IL-6, and IL-1, immune dysregulation with M1 macrophage polarization, and increased Th17 [4]. Finally, apical migration of the covering epithelium may occur, leading to progressive bone exposure. The molecular pathways involved in such occurrence mainly consist of the increased production of type III collagen, MMP-9, and MMP-13 levels. Moreover, the presence of cells positive for a-SMA is suggestive for myofibroblast presence [5].

Several risk factors have been clearly identified as contributors to disease onset and development, such as diabetes, smoking habits, trauma related to the use of removable dental prostheses, inadequate oral hygiene, and a recent history of invasive (tooth extraction, implant insertion, oral surgery procedures (especially for bone lesions, etc.) and noninvasive (such as scaling, endodontic treatment, etc.) dental procedures [6]. Therefore, to determine the overall risk of MRONJ occurrence and to give a customized therapy for each patient, it is fundamental to assess the comorbidities associated with the patient’s clinical history [7]. The incidence of MRONJ has been reported to be considerably lower in non-oncologic (e.g., osteoporotic patients and patients with non-malignant bone diseases) compared to oncologic patients due to the different therapeutical schemes of administration. Oncologic patients require higher doses and more infusions of medications, resulting in massive cumulative doses [8]. The drugs that affect bone remodeling are derived from several families. They show a direct involvement in the development of MRONJ: since 2003, when Marx first described bisphosphonate-related osteonecrosis of the jaws (BRONJ), the number of reported cases has risen globally, and the development and introduction of new drugs related to MRONJ, such as antiangiogenetic agents or antiresorptive medications, make it difficult to estimate the actual entity of MRONJ accurately [9,10]. Therefore, a wider range of newer medications have been introduced recently in the therapy of oncologic and non-oncologic patients and have also been associated with a possible risk of MRONJ, such as tyrosine kinase inhibitors (TKIs), monoclonal antibodies, mammalian target of rapamycin inhibitors, selective estrogen receptor modulators, and immunosuppressants [9]. In the literature, the number of cases of MRONJ caused by new medications is limited to case reports or case series, leading to a lack of consensus about the determination of the risk degree and the management of these patients [11]. Within a constantly growing list of medications possibly related to MRONJ occurrence, Osimertinib, a third-generation epidermal growth factor receptor (EGFR) tyrosine kinase inhibitor (TKI) that is highly selective for EGFR-activating mutations and the EGFR T790M mutation, has been recently suspected to be a potential one [12]. It is generally administered for the standard treatment of patients with previously untreated *EGFR*-mutated advanced non-small-cell lung cancer (NSCLC) [13]. The Food and Drug Administration (FDA) approved Osimertinib in 2018, and the most common adverse reactions described are diarrhea, rash, dry skin, nail toxicity, and fatigue [6]. Moreover, in patients who reduce the dose or interrupt the regimen, the most frequent complications are the prolongation of the QT interval upon electrocardiogram (ECG) examination, neutropenia, and diarrhea [14,15]. MRONJ is not mentioned in their leaflet’s adverse reactions section, although the TKI family has been linked to MRONJ [11]. At the moment, there is only one study that reports an MRONJ case related to Osimertinib therapy [12]. After these relevant considerations, the prevention, diagnosis, and management of MRONJ require a multidisciplinary approach involving oncologists, dentists, and maxillofacial surgeons [16], the latter usually being the first clinicians to identify patients. Diagnosis and staging remain the pivotal factors in MRONJ patient management.

In the current study, we report an NSCLC patient showing two maxillary stage 2 MRONJs after being subjected to oral surgery during Osimertinib therapy, thus suggesting a possible Osimertinib-related origin. Moreover, we conducted a review to investigate the latest literature about Osimertinib-related MRONJ.

## 2. Case Report

The clinical case reported herein is described according to the CARE guidelines. Written informed consent was obtained from the patient for the diagnostic and therapeutic procedures and the possible use of their data for scientific purposes.

### 2.1. Patient Information

A 70-year-old male patient was referred to the Complex Unit of Odontostomatology of the University of Bari “Aldo Moro” complaining of two bone exposures in the upper maxillary in the first and second quadrant, draining purulent exudate. The patient’s medical history revealed that, in 2018, he received a diagnosis of non-small-cell lung cancer (NSCSL) EGFR+ with multiple bone metastases, treated with Osimertinib from 2020 and Denosumab (Xgeva 120 mg IM; one administration every 28 days) from October 2020 to February 2022.

### 2.2. Timeline

Denosumab was suspended for 5 months before multiple teeth extractions. Osimertinib was never suspended. As refered by the patient’s general dentist, from July 2022 onwards, all teeth were progressively removed. So, after 2 months, which allowed complete mucosal healing, two removable dentures were delivered to the patient in September 2022. The patient reported the appearance of the maxillary lesions in May 2023, ending (to our knowledge) in June 2023. Thus, the Denosumab-free period lasted 10 months from teeth extraction to bone exposure and 15 months from drug suspension to bone exposure. The medical history timeline is illustrated in Figure 1.

### 2.3. Clinical Findings and Diagnostic Assessment

The intraoral examination revealed bilateral maxillary bone exposure, draining purulent exudate through a fistula, responsive to NSAIDs. The right side (I quadrant) presented a swelling with a small clinical diameter (approximately 2 cm × 2 cm) bone exposure. However, it showed deep probing. The left side (II quadrant) had a wider clinical diameter (approximately 2 cm× 1.4 cm) of bone exposure, also with deep probing. The initial clinical situation is reported in Figure 2.

With the suspicion of MRONJ, the patient underwent a panoramic radiogram and computed tomography (CT scan) to better define the dimensions of necrotic areas and perform proper staging. The panoramic radiogram showed osteolytic lesions with ill-defined borders surrounding necrotic bone both at the sites of 1.6 and 2.3. This was confirmed by CT scans, which provided more details (Figure 3).

After CT scans, the clinicians attributed this case to the II stage of the Favia et al. classification [17], which also corresponds to the II stage of the latest Ruggiero et al. position paper [1], because of exposed and necrotic bone, bone probing, purulent exudate draining, and CT findings in a symptomatic patient responsive to NSAID.

### 2.4. Therapeutic Intervention

Based on the guidelines mentioned above, the patient was first subjected to non-operative therapy, starting 3 cycles of therapy with Ceftriaxone IM (1 g daily) and 250 mg of metronidazole twice a day per os for 7 days, with a 10-day drug-free period between each cycle, together with a local antimicrobial rinse (Chlorhexidine 0.2%). After the second cycle of medical therapy, the right maxillary swelling apparently reduced, while the exposed bone on the left was wider. A new panoramic radiogram and CT scan showed a progressive bone sequestrum with a better definition margin, thus suggesting a partial healing with bone sequestrum formation, giving more evidence of MRONJ probably related to Osimertinib therapy (Figure 4).

The patient was further visited after the third antibiotic therapy and a free drug period of 30 days in order to evaluate the necessity and possibility of surgical debridement, mainly for diagnostic purposes aimed to exclude secondary metastatic lesions; obviously, with the suspicion of an MRONJ related to a drug (Osimertinib) actually taken by the patient, he was accurately informed about the possible risk of further complications. So, a surgical bone debridement (classifiable as a noninvasive surgical procedure) under local anesthesia was performed to remove the necrotic bone of both lesions for the following histological examination (Figure 5).

Histopathologic examination confirmed the clinical suspicion of bilateral maxillary osteonecrosis with the following diagnosis: “Mucosal fragments showing erosion and reactive pseudo-papillary hyperplasia of the epithelium, with extensive inflammation in the lamina propria. Osseous fragments show rare hematopoietic marrow, large osteons with rarefied Haversian channels, and a lack of vascular vessels. Also, inflammatory infiltration and granulocytic neutrophilia, necrotic bone and bacterial colonies are present in the sample. Fibrotic stroma was composed of aggregates of lympho-plasma cell infiltrates and hemosiderin deposition. The diagnosis was compatible with Medication-Related Osteonecrosis of the Jaws.” (Figure 6).

### 2.5. Follow-Up

Complete healing occurred 6 weeks after the surgical time. The clinical follow-up is reported in Figure 7.

## 3. Discussion

The current study aims to report an infrequent case of MRONJ, probably related to Osimertinib administration in a patient with a diagnosis of pulmonary cancer, evaluating the current literature too.

### 3.1. General Definition of MRONJ

At the first description in 2003 by Marx et al. [18], MRONJ was called bisphosphonate-related osteonecrosis of the jaws (BRONJ), defined by the authors as “*painful bone exposure in the mandible, maxilla, or both, that were unresponsive to surgical or medical treatments in patients taking Bisphosphonates*” [18]. After 11 years, the denomination of BRONJ changed in MRONJ because further medications were added related to jaw osteonecrosis: in 2014, the position paper from the AAOMS implemented bisphosphonates in patients who began antiresorptive therapy, such as RANK-L inhibitors, antiangiogenetic therapy, vascular endothelial growth factor (VEGF) inhibitors, and tyrosine kinase inhibitors [19]. In the updated position paper from the AAOMS published in 2022, fusion proteins (aflibercept), mTOR inhibitors (e.g., Everolimus), radiopharmaceuticals (radium 223), selective estrogen receptor modulators (raloxifene), and immunosuppressants (methotrexate and corticosteroids) were also included [1]. As for MRONJ staging, the four levels suggested by Ruggiero et al., 2022 [1] should be used to distinguish them as such:Stage 0: patients with specific clinical symptoms or radiographic findings without clinical evidence of necrotic bone;Stage 1: Exposed and necrotic bone or fistula without inflammation;Stage 2: Exposed and necrotic bone or fistula with inflammation;Stage 3: Exposed and necrotic bone or fistula with signs of infection and variably exposed necrotic bone extending beyond the alveolar bone region, pathologic fracture, extraoral fistula, oral antral/oral–nasal communication, and osteolysis extending to the inferior boundary of the mandible or sinus floor.

### 3.2. Systemic Diseases Associated with MRONJ Occurrence

It is essential to highlight that patients beginning to take these medications generally belong to three populations: cancer patients, osteoporotic patients, and patients with non-malignant bone diseases (such as Paget disease), presenting different risk degrees. The drug class, their administration (oral, subcutaneous, IM, or EV), their posology, the duration, and the possible combinations between multiple drug therapies profoundly affect the risk of developing MRONJ [2]. Considering the wide range of medications and the increasing number of patients receiving these therapies, MRONJ incidence is increasing yearly, up to 1231.7 per 100,000 person-years [20]. The most frequent disease needing antiresorptive therapy is postmenopausal osteoporosis, which requires minor drug circulating levels (as oral or monthly/weekly IM administration). But, considering the absolute higher number of osteoporotic patients compared to cancer patients, a significant proportion of overall MRONJ cases are represented by osteoporotic patients [21]. But, at the moment, the prevalence rate of MRONJ is considerably higher in cancer patients that need more circulating levels of antiresorptive drugs to control diffuse metastases [22]. The cancers mostly associated with this occurrence are breast cancer for women, prostate cancer for men, and multiple myeloma for both [23], which are also the most frequent cancers harboring metastases to the bones [24]. The third population of patients exposed to MRONJ risk is affected by non-malignant bone diseases, which are relatively rare. In fact, patients suffering from giant cell bone tumors, osteogenesis imperfecta, and Paget disease benefit from antiresorptive therapy that gives them an estimated MRONJ risk of up to 5% of the prevalence rate of the overall low number of cases [25]. Along with the primary indication of antiresorptive therapy, the patient’s comorbidities must always be carefully considered: diabetes mellitus, rheumatoid arthritis, and all the immunocompromising pathologies or related therapies are generally reported to be associated with a higher risk of MRONJ development [26].

### 3.3. The Etiopathogenetic Mechanisms of MRONJ

There are many etiopathogenetic theories in this regard. As reported by Lombard et al. [27], we can distinguish two main theories: the “inside-outside” and “outside-inside” mechanisms. The “inside-outside mechanism” consists of bone-reduced turnover, osteoclast inhibition, and reduced vascularity induced by antiresorptive therapies, which leads to bone necrosis that is secondarily exposed to microorganism infection, which may explain “spontaneous necrosis” [28]. On the other hand, the “outside-inside” theory suggests that local immune dysregulation, triggered by chronic infective diseases such as periodontitis or surgical trauma, is spread to the bone, which, due to its reduced capabilities of resorption and remodeling, results in osteocyte necrosis. This theory could explain surgically triggered or periodontal osteonecrosis [29]. However, the immune system has been linked to the pathogenesis of MRONJ as BPs cause immune dysregulation that leads to the expression of pro-inflammatory molecules; they increase the production of IFN-γ, suppress the differentiation of dendritic cells, and increase the chemotaxis, phagocytosis, and oxidative stress of neutrophil granulocytes [30]. Besides etiological theories, the literature describes how antiresorptive therapies affect bone remodeling. BPs bind to the bone surface and, during resorption, are released, resulting in osteoclast apoptosis and a stop to bone resorption [31]. On the other side, BPs reduce the osteogenic response to osteoblasts [32]. Besides bone remodeling, BPs and non-anti-resorptive medications affect bone angiogenesis. These drugs inhibit VEGF, TNFα, and the production of pro-angiogenic cytokines such as IL-17, resulting in a progressive vascularity decrease that could lead to avascular necrosis [33]. However, patients in monotherapy with antiangiogenic drugs present a minor risk compared to those on antiresorptive therapy [34]. MRONJ could be spontaneously or surgically triggered; principally, tooth extraction and any surgical treatment (such as implant insertion) performed on patients taking one or more of the above-cited medications are associated with a risk of 14.1% among cancer patients and up to 1% among osteoporotic patients [22,35].

### 3.4. Updated Drugs Related to MRONJ

Due to the well-known action mechanism of bone remodeling, the drugs first described to be associated with ONJ were bisphosphonates [18]. Then, after BPs, the most strongly associated drug is Denosumab. Both are discussed in the previous section. Furthermore, there are several new drugs associated with the MRONJ risk. They are considered non-antiresorptive medications and include several drug families [36]; they are summarized in the following table (Table 1).

It is evident that the main problem that clinicians must face is the continuously updating list of new therapies (such as monoclonal antibodies, TKIs, or biologically targeted therapies) associated with a risk of MRONJ.

### 3.5. Osimertinib and MRONJ

As for Osimertinib, it is an irreversible third-generation epidermal growth factor receptor (EGFR) tyrosine kinase inhibitor (TKI) approved by the FDA and EMA as therapy for metastatic EGFR-mutant non-small-cell lung cancer (NSCLC) patients that have acquired the EGFR T790M resistance mutation [52,53]. When epidermal growth factor (EGF) binds to the extracellular domain, it induces not only EGFR homo-dimerization but also hetero-dimerization with the other members of the ErbB family of receptor tyrosine kinases such as Her2 (ErbB2), Her3 (ErbB3), and Her4 (ErbB4). Consequently, receptor dimerization determines the activation of the receptor kinase, inducing intracellular signal cascades, which play a significant role in epithelial tissue maintenance and growth [54,55]. Lung cancer is the second most diagnosed cancer and is highly associated with cancer-related deaths every year [56], with more than a 50% rate of bone metastasis during the disease course [57], so an association between the medical targeted therapy of NSCLC and antiresorptive therapy is common. Histologically, 81% of lung cancers are classified as NSCLC, including lung adenocarcinoma, squamous cell carcinoma, and large-cell carcinoma [58]. NSCLC consists of a group of different diseases that can present a mutation of epidermal growth factor receptor (EGFR), an oncogenic driver, which can be used as a target for developing new drugs for cancer-targeted treatment [54]. The leading role of Osimertinib is due to the specific competitive inhibition of intracellular pathways, which is guaranteed by irreversible bonding with EGFR, avoiding cellular replications and growth. As for its effectiveness, Osimertinib is considered the first-line therapy for untreated EGFR mutation-positive advanced NSCLC and is well tolerated by cancer patients [13]. The most common adverse reactions described occurring in patients in treatment with Osimertinib are rashes, diarrhea, vomiting, dry skin, nail toxicity, stomatitis, and fatigue.

Furthermore, patients who had a dose reduction or a dose interruption presented a prolongation of the QT interval at the ECG examination, neutropenia, and diarrhea; however, MRONJ was not listed [14]. Bone involvement during Osimertinib therapy was explained in an interesting study by Kanaoka et al. about bone metastasis in 45 NSCLC patients during Osimertinib therapy, evaluating their CT images [59]. They found that 82% of these patients developed an osteoblastic bone reaction (OBR) in their bone metastasis. Other studies report similar results after first-generation TKI drugs, but the percentage of the OBR during these therapies was inferior to Osimertinib therapy. Since 2010, Ansen et al. have reported the first three cases of OBRs during NSCLC harboring the EGFR mutation in treatment with first-generation TKIs [60]. Yamashita et al. reported that 27% of patients subjected to Gefitinib showed OBRs [61], and Pluquet found that 77% of patients in therapy with Erlotinib or Gefitinib had ascertained OBRs [62]. Bersanelli et al. reported OBRs in 27% of their patients during first-generation TKIs treatment [63]. It is evident that the percentages of patients developing OBRs during the first-generation TKI treatment are not consistent between them. Still, all the authors agree about the clinical significance of this occurrence. In fact, the bone remodeling of diffuse metastasis in an osteosclerotic manner is considered disease control and a good outcome of TKI medical therapy. Moreover, Osimertinib, which represents the actual first-line treatment of NSCLC carcinoma, also shows better results regarding OBRs for diffuse bone metastasis.

### 3.6. Case Discussion

The present case report should also be considered a warning in terms of the possible risk of MRONJ occurrence, both for oncologists who prescribe Osimertinib and for oral and maxillo-facial surgeons [1]. A bilateral maxillary II stage necrosis was reported during therapy with Osimertinib. The Denosumab drug holiday in cancer patients is still under debate, and there is no general consensus about their management prior to any surgical procedure [64,65,66]. Denosumab’s mean half-life is 25.4 days because RANK-L inhibitors do not bind bone, unlike bisphosphonates. Non-significant plasmatic levels of Denosumab are found after 4 months [67], so this time could indeed represent a sufficient drug-free period to prevent tooth extraction-related necrosis (in surgically triggered MRONJ). Nevertheless, there are some studies that do not suggest a Denosumab drug holiday in cancer patients because of the increased risk of disease progression [64,65,68]. On the other hand, there is a case series that suggests Denosumab suspension 12 months prior to any surgical procedures [69]. Thus, the authors considered the 2022 AAOMS position paper as guidelines, which suggests a Denosumab drug holiday of 3–4 months, due to the insignificant levels of osteoclast inhibition after that period [1]. In particular, the onset of this maxillary MRONJ was 15 months after Denosumab suspension, so we believe that Denosumab can be excluded as the etiopathogenetic cause, which is mostly related to the exclusive Osimertinib regimen or eventually a drug combination of Denosumab (albeit suspended following the AAOMS’s suggestions) and Osimertinib. The etiopathogenetic cause of MRONJ in the current case is still unknown because it could either be surgically triggered or removable prosthesis-triggered. Nevertheless, the patient underwent tooth extractions and removable-denture application both in the upper and lower arches but developed MRONJ only in the upper, so attributing it to a single cause is challenging. Such a consideration arises as, besides teeth extractions (considered the most common risk factor), removable denture-related trauma could certainly be related to MRONJ occurrence, as already reported in the literature [1,70,71]. After this, to the best of our knowledge, MRONJ during Osimertinib therapy has only been described in one similar case report by Wang et al. in 2022 [9], who reported a maxillary MRONJ after tooth extraction in a 69-year-old female patient affected by NSCLC, who had been taking Osimertinib for the last 4 years. Then, we reported our bilateral stage 2 MRONJ during the same therapy. Overall, there are three cases of MRONJ during anti-EGFR therapy, sustained by the timing and the anti-vascular and bone alteration properties of these drugs, which we exposed previously. The rare occurrence, however, does not exclude the possibility of MRONJ developing in these patients. Consequently, the lack of data in the literature about it leads there being no evidence of any consensus about these patients’ management or the need for an Osimertinib holiday period. Usually, in the management of patients at risk of MRONJ, a drug holiday is always suggested prior to surgical procedures, but it must be conducted according to the oncologist. However, in this case, the patient was not allowed to suspend Osimertinib by the oncologist, because the risk of cancer progression during their suspension was higher than the benefits, and there are no data available in the current literature about the management of MRONJ during Osimertinib therapy. Additionally, as for therapy, our report provides evidence that a conservative surgery, preceded by the conventional medical therapy (three cycles therapy with Ceftriaxone (1 g daily, IM) and metronidazole (250 mg, twice a day per os for 7 days), with a 10-day drug-free period between each cycle, together with a local antimicrobial rinse of 0.2% Chlorhexidine), was resolutive. It is important to underline this, as a consensus about the therapeutical management of MRONJ, which is generally classified as conservative (non-operative) and surgical, is still lacking. Also, the same antibiotic use (type, dosage, and duration) in MRONJ patients lacks evidence; generally, a broad-spectrum antibiotic, mainly a combination of amoxicillin/clavulanic acid (effective against Gram-negative and β-lactamase-resistant Gram-positive bacteria) and metronidazole (effective against Gram-positive cocci and anaerobic), is suggested for infection control along with antiseptic rinses [1]. On the basis of our experience, a medical therapy as performed in the current case (the combination of Ceftriaxone and metronidazole) provides excellent control of infection in the pre-operative stage, leading to a good mucosal inflammation reduction (mandatory for closure of the surgical site) along with perilesional bone remodeling with necrosis delimitation (helpful to limit perilesional bone extension during surgery) [6]. As said before, our case is consistent with Wang et al.’s report and supports their hypothesis. Considering that NSCLC represents 81% of all lung cancers and lung cancer is the second most diagnosed cancer worldwide [72], Osimertinib-related MRONJ cases are expected to increase in incidence. This alert must be spread and shared with dentists and oncologists in order to discuss together the management of these patients to possibly determine an Osimertinib drug holiday before surgical treatments of jaws, considering that there are no data about TKI suspensions before oral surgery to prevent MRONJ. Moreover, we suggest adding Osimertinib to the MRONJ-related drug list in order to follow the surgical procedures reported by the AAOMS that are useful to reduce the risk of MRONJ, such as primary wound closure, preventive antibiotic administration, and adjunctive therapies such as low-level laser therapy and chlorhexidine mouthwash [1]. There are several limitations of this study: the combination with Denosumab could be considered a bias, and the unknown etiopathogenetic cause (surgical- or denture-triggered) and lack of studies that directly relate MRONJ to Osimertinib therapy make it impossible to draw firm conclusions. Further studies are needed to confirm the hypothesis that Osimertinib monotherapy can cause MRONJ. Finally, a routine oral examination and monitoring should be advised and suggested in these patients, especially when oral surgery procedures should be planned.

## 4. Conclusions

The case herein described report on a new medication (Osimertinib) possible related to MRONJ occurrence in lung cancer patient, although the patient had previously undergone therapy with Denosumab but suspended before teeth removal; also, the wearing of the removable prosthesis after teeth removal represents a further element to consider carefully. Nevertheless, similar situations are frequent in the clinical practice and theoretically their incidence could increase over time in proportion to the ever-increasing diffusion of such therapies (e.g., Osimertinib) mostly in lung cancer patients. Therefore, the reporting of similar case surely is important for the scientific community along with their clinical and/or surgical management which, anyway, is strictly related to the patients’ health.

## Figures and Tables

**Figure 1 healthcare-12-00457-f001:**
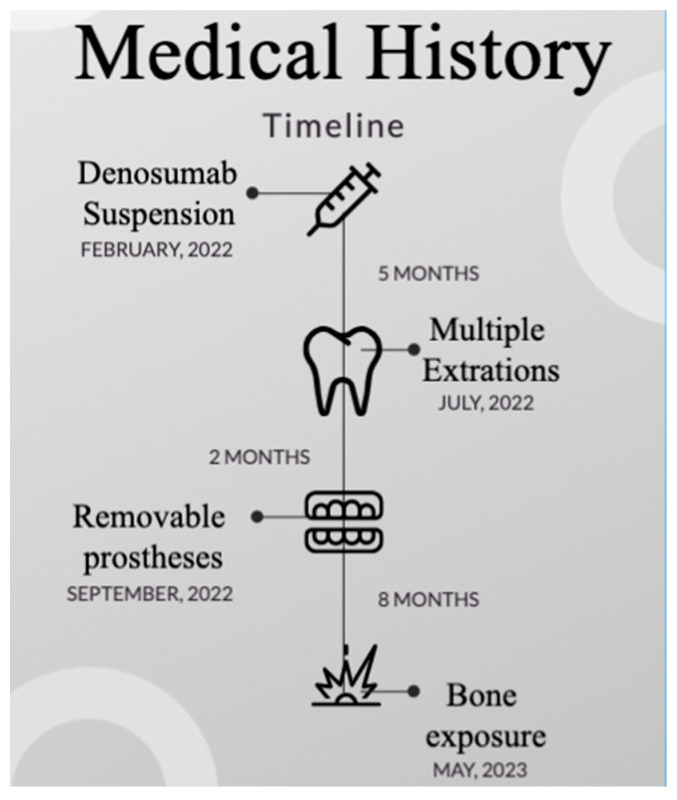
Medical history timeline. Osimertinib was not suspended.

**Figure 2 healthcare-12-00457-f002:**
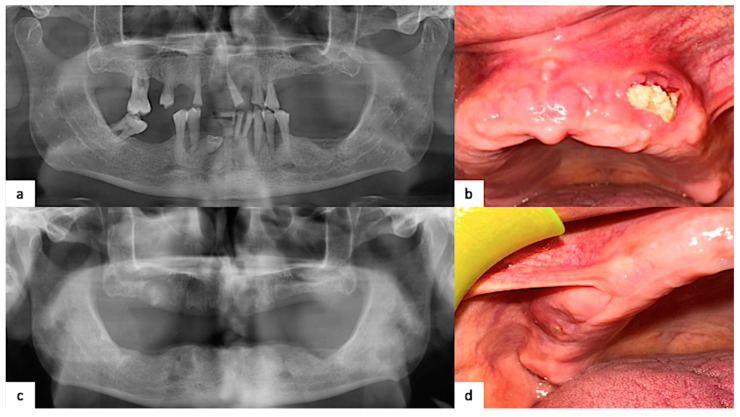
(**a**) Orthopantomography of the jaws before multiple teeth extractions; (**b**) at the time of our observation, bone exposure in the left maxilla; (**c**) first panoramic radiogram with evidence of an adjunctive osteolytic lesion at the site of 1.6; (**d**) right maxillary swelling with small draining bone exposure.

**Figure 3 healthcare-12-00457-f003:**
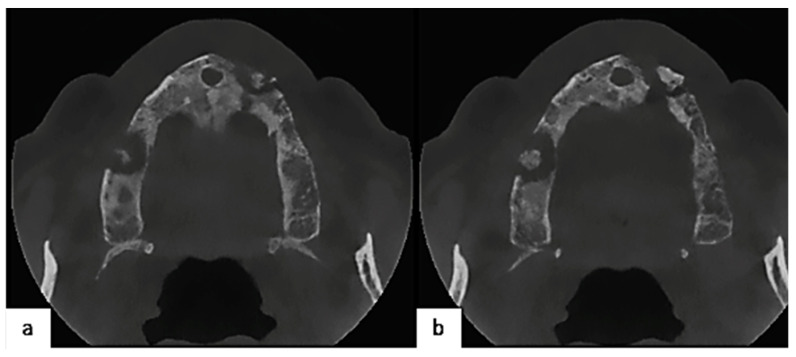
CT axial scans better show the osteolytic lesions on both sides of the maxilla. (**a**) Cranial slice; (**b**) caudal slice.

**Figure 4 healthcare-12-00457-f004:**
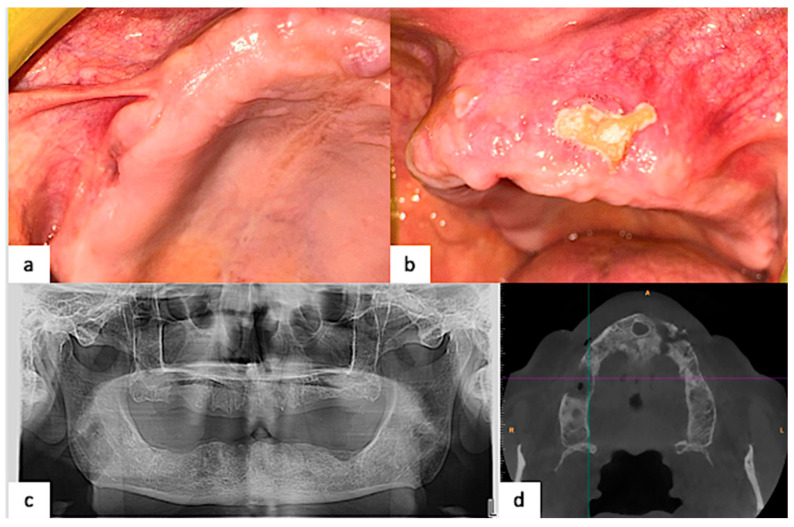
Evident reduction in the lesion of the right maxilla and persistency of exposed bone in the left after 2 cycles of medical therapy. (**a**) Right maxilla; (**b**) left maxilla; (**c**) panoramic radiogram showing progressive bone sequestrum with a better definition of lesion border; (**d**) CT scan showing bone sequestrum area with an ill definition of both lesions.

**Figure 5 healthcare-12-00457-f005:**
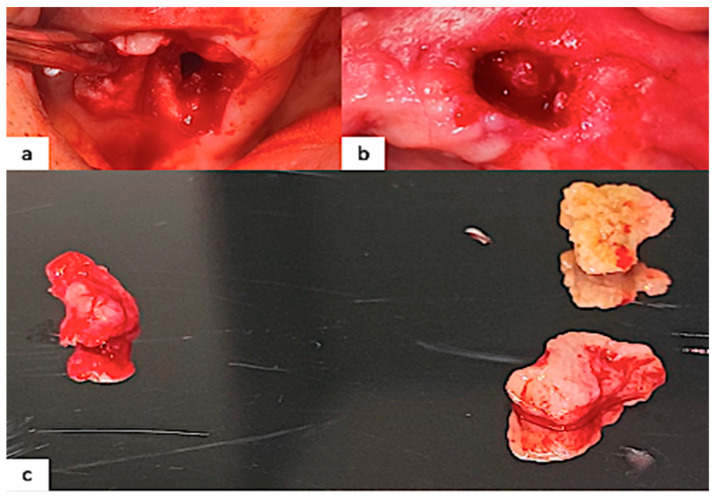
(**a**) Immediate postoperative view after surgical bone debridement in the left maxilla; (**b**) surgical site in the right maxilla where a bone debridement was performed; (**c**) samples of necrotic bone.

**Figure 6 healthcare-12-00457-f006:**
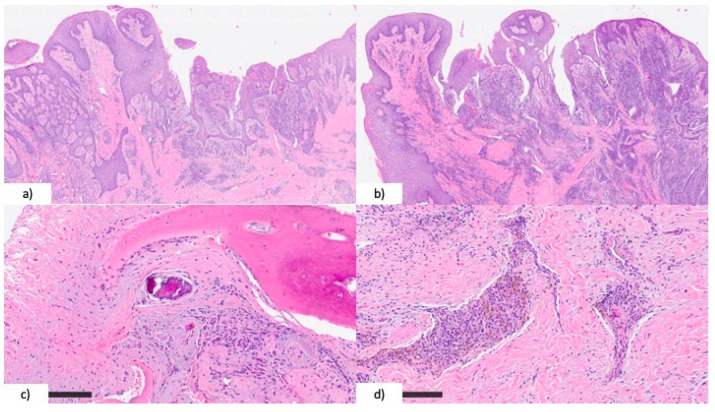
(**a**) Oral mucosa showing erosion and reactive pseudo-papillary hyperplasia of the epithelium (H&E, original magnification ×2); (**b**) conspicuous and extensive inflammation was present in the lamina propria (H&E, original magnification ×2); (**c**) closer detail revealing the histological appearance of bone necrosis (H&E, original magnification ×10); (**d**) fibrotic stoma with aggregates of lympho-plasma cell infiltrates and hemosiderin deposition (H&E, original magnification ×10).

**Figure 7 healthcare-12-00457-f007:**
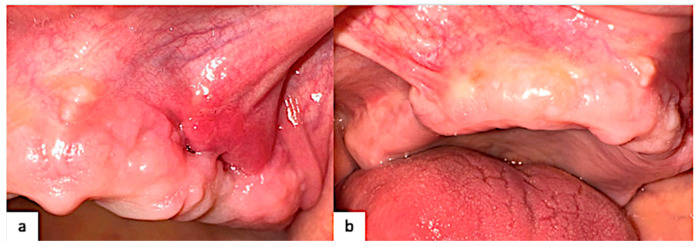
Healing after 6 weeks shows a normal pink-red gingiva (**a**) Left maxillary; (**b**) right maxillary.

**Table 1 healthcare-12-00457-t001:** Drugs possibly related to MRONJ occurrence.

Drug Type	Drug Family	Drug Target	Active Substance
Antiresorptive	Bisphosphonates	Osteoclasts	Alendronate, Ibandronate, Neridronate, Pamidronate, Risedronate, Zoledronate, Clodronate, Etidronate, Tiludronate [6]
	Monoclonal Antibodies	RANK-L	Denosumab [37]
Non-antiresorptive	Antiangiogenic Monoclonal Antibodies	VEGF-A	Bevacizumab [38]
	T Cytotoxic lymphocyte activator	CTLA-4	Ipilimumab [39]
	Monoclonal Antibodies	CD-20	Rituximab [40]
	Monoclonal Antibodies	HER2/neu	Trastuzumab [41]
	Monoclonal Antibodies	HER2/neu	Pertuzumab [41]
	Antiangiogenic Monoclonal Antibodies	VEGF-A	Ranibizumab [42]
	TKI	PDGFR and VEGFR	Sunitinib [43]
	TKI	ABL, C-KIT, and PDGF-R	Imatinib [44]
	TKI	c-Met, VEGFR2, AXL, and RET	Cabozantinib [45]
	TKI	VEGFR and PDGFR	Sorafenib [46]
	TKI	EGFR	Erlotinib [47]
	TKI	EGFR	Gefitinib [48]
	TKI	EGFR	Osimertinib [12]
	mTOR inhibitor	mTORC1	Everolimus [49]
	Selective Estrogen Receptor Modulators	Estrogens	Raloxifene [50]
	Recombinant Protein	VEGF-A and VEGF-B	Aflibercept [51]

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
