# Peer review of "Could MRONJ Be Related to Osimertinib Monotherapy in Lung Cancer Patients after Denosumab Suspension?"

_healthcare, 2024, doi:10.3390/healthcare12040457_

Round 1
Reviewer 1 Report
Comments and Suggestions for Authors
This is a relevant study in which the Authors present a case report suggestive of Medication-related Osteonecrosis of the Jaws (MRONJ) possibly related to Osimertinib therapy and address this issue according to studies in the literature.
Some more relevant considerations:
- Although the mechanisms involved in the pathogenesis of Medication-related Osteonecrosis of the Jaws (MRONJ) are not completely elucidated and there is still much to be researched, several factors and molecular events have been related to the pathogenesis of MRONJ and it is pertinent that they are mentioned more clearly in the introductory text and are taken up in the discussion for a broader and more detailed approach, considering data available in the literature especially on aspects related to the physiology and remodeling of bone tissue, vascularization, immune system, association with other conditions or systemic diseases, drug interactions, impact of these classes of medications on biological processes and signaling pathways that can affect jaw bone homeostasis, procedures and dental interventions, pharmacology and toxicology of medications potentially related to MRONJ, including Osimertinib;
Additional considerations:
- I suggest that the Authors include a table with the most relevant studies in the literature that associate the different classes of medications with MRONJ;
- Some paragraphs could be better organized especially in the discussion text (very short paragraphs intercalated with very long paragraphs).
- Resize the figures inserted in the manuscript for better standardization, if applicable.
Reviewer 2 Report
Comments and Suggestions for Authors
This case report describes a case of MRONJ onset due to a third-15 generation epidermal growth factor receptor tyrosine kinase inhibitor (Osimertinib). His medical history revealed a two-year Denosumab course along with Osimertinib. While the patient had discontinued Denosumab, Osimertinib was not stopped prior to extractions. This resulted in MRONJ, confirmed histologically. The clinicians performed a sequestrum removal and bone debridement after three cycles of antibiotic therapy.
Comments:
1. The study should be labelled as a case report, instead of a review.
2. The introduction is too long. For example, Line 91-106 is unnecessary, and could be in the discussion instead.
3. Was the Osimertinib suspended after discovery of the possible MRONJ?
4. I am not totally convinced that the study shows a causation between Osimertinib and MRONJ. A) The time of denosumab suspension to multiple extractions was 5 months only. It is not known if MRONJ had already developed by then during denture fabrication. B) The dentures may have been a contributory factor due to poor fabrication resulting in traumatic ulcers. C) Although the half life of denosumab is 25.4 days, there is still a residual level of denosumab. Even the authors mention that denosumab drug holiday is under debate, and the 15 months ‘onset’ (if it really only had an onset at 15 months to begin with) is totally the authors’ opinion only. D) they may have been interactions between Osimertinib and Denosumab which resulted in MRONJ, so perhaps the title/aim should be changed to possible interactions between these two drugs, rather than stating that Osimertinib is a causative factor.
5. The discussion is excessively long without proper structure.
6. As a case report – please provide ethics review, documentation of reasons (if so) why it was exempt from review, patient consent form. The report should also follow CARE guidelines, and the authors should attach the checklist here.
Comments on the Quality of English LanguageFigure 1 - 'Denosumab Sospension'. Please correct to suspension.
Reviewer 3 Report
Comments and Suggestions for Authors
Thank you for the opportunity to consider this interesting work.
The manuscript submitted for potential publication in Healthcare reports about a third-generation epidermal growth factor receptor tyrosine kinase inhibitor (Osimertinib) as possibly responsible of bilateral maxillary necrosis onset in the herein described case.
Dear Authors,
Presented manuscript needs some additional revisions.
The paper is very nicely written.
Unfortunately, it would be difficult to sustain the research thesis.
As you mentioned, teeth extractions and removable dentures-related trauma could be surely related to MRONJ occurrence, so I suggest removing the thesis that Osimertinib is responsible for it.
Also, the discussion should be reorganized. It is too long.
Title
The title is difficult to read because of many abbreviations. And because the thesis could not be supported by your findings, please rewrite it.
Introduction
Line 40 „MRONJ has been clinically defined by the ‘’American Association..” In the next sentence, you repeat the definition. Please remove it, the definition is clear.
Line 54 „ to disease onset and development (such as diabetes, smoking..”- the bracket is missing
Line 100 „ Stage 0 (patients with a-specific clinica”…- the bracket is missing
Discussion
Line 223 and following till the end of paragraph: „For example, Guida et al. exposed in their recent report, as many new medications have been added to the list of potentially associated with MRONJ development, a case of a patient that presented MRONJ although she suspended the therapy with ipilimumab (a monoclonal antibody directed against CTLA4 receptors) three years before for metastatic melanoma...”- please rewrite these sentences, they are not clear.
Line 234 „Denosumab's mean half-life is 25.4..” – too long sentence.
Line 276 „ it is relevant to consider that the latter could affect bone metabolism in the same way as Bisphosphonate”- It does not seem that there is a proof for such a thesis. Please, remove it. As you write later, the rare occurrence, however, doesn’t exclude the possibility of developing MRONJ in patients, but it is not possible to draw such conclusions based on a few cases.
Line 321” by Mohamed et al. in a patient in a 53-year-old female treated..”-rewrite it please, it is not clear
Line 353 „As said before, our case is consistent with Wang et al. report, further supporting their hypothesis to add Osimertinib to the list of drugs related to ONJ, also reporting such risk on the drug package leaflet.”- I would suggest not to follow the case of one patient to formulate the general rule. Please, remove this aspect of the paper.
Round 2
Reviewer 1 Report
Comments and Suggestions for Authors
The adjustments complemented and improved the manuscript.
Author Response
Thank you for your comments.
Reviewer 2 Report
Comments and Suggestions for Authors
Thank you for the changes.
I originally requested the paper to be written as a case report but the authors have mentioned that the paper meets prerequisites to be classified as a review. While the authors have termed this as a review (MAY MRONJ BE RELATED TO OSIMERTINIB MONOTHERAPY IN LUNG CANCER PATIENT AFTER DENOSUMAB SUSPENSION?), I am concerned how much of the review is really relevant to the objective of the study ('to investigate the latest literature about Osimertinib-related MRONJ') and the title. The discussion only identified one other paper that reports this incidence. Much of the discussion explains MRONJ in general, and even the table covers drugs (non-anti resorptives), and not osimertinib. Only the paragraph 3.5 ('Osimertinib and MRONJ') is truly relevant to the review, which takes up a relatively small space.
If this was a review on non anti-resorptives (of which the review appears so), then it would require a change in the title, the objectives, and the case report may be less relevant as well). It will also beg the question of how novel this review is (see for example PMID: 35013781). I am of the view that this paper should be written as a case report, and much of the report focusing only on the case (condense most of the discussion and remove irrelevant sections to the case report).
Comments on the Quality of English LanguageCan be improved further. For example, in the title, MAY MRONJ BE RELATED TO OSIMERTINIB MONOTHERAPY IN LUNG CANCER PATIENT AFTER DENOSUMAB 3 SUSPENSION?
PATIENT should be written as 'patients'.
Reviewer 3 Report
Comments and Suggestions for Authors
Thank you for your hard work.
Author Response
Thank you for your comments.